# Cardio-Respiratory Events and Food Autonomy Responses to Early Uni-Modal Orofacial Stimulation in Very Premature Babies: A Randomized, Controlled Study

**DOI:** 10.3390/children8121188

**Published:** 2021-12-16

**Authors:** Sahra Méziane, Véronique Brévaut-Malaty, Aurélie Garbi, Muriel Busuttil, Gaelle Sorin, Barthélémy Tosello, Catherine Gire

**Affiliations:** 1Department of Neonatology, North Hospital, APHM, Chemin des Bourrely, 13015 Marseille, France; sahra.meziane@ap-hm.fr (S.M.); veronique.brevaut@ap-hm.fr (V.B.-M.); aurelie.garbi@ap-hm.fr (A.G.); tosellobarthelemy@hotmail.fr (M.B.); gaelle.sorin@ap-hm.fr (G.S.); catherine.gire@ap-hm.fr (C.G.); 2CNRS, EFS, ADES, Aix-Marseille University, 13344 Marseille, France; 3CEReSS-Health Service Research and Quality of Life Center, Aix-Marseille University, 27 Boulevard Jean Moulin, 13005 Marseille, France

**Keywords:** very preterm, cardiorespiratory events, orofacial stimulation, autonomy

## Abstract

Uni-modal orofacial stimulation (OFS) for preventing very preterm infants’ oral disorders is highly controversial. Our study sought to demonstrate that OFS reduced cardio-respiratory events and improved food autonomy in a population of very preterm infants. Our study was randomized, controlled, prospective, and unicentric. The preterm included were born between 26–29 weeks gestational age (GA) with a corrected postnatal age <33 weeks GA. They were randomized into two groups: the experimental group underwent OFS, according to a protocol established, over 10 consecutive days, and the control group underwent no OFS. The primary outcome was the number of cardiorespiratory events: apnea–bradycardia (with or without desaturations) or number of isolated desaturations, which were evaluated at four separate times. Measurements occurred during the first, fourth and eighth independent feedings. Seventeen patients were included in the experimental group and 18 in the control group. The number of cardiorespiratory events for all independent feeding times was significantly reduced in the OFS group (*p* = 0.003) with univariate analysis, but not with multivariable analysis. The quantity of milk ingested during the first autonomous feeding was higher in the experimental group. The acquisition of food autonomy and the duration of hospitalization were similar in the two groups. While our study does not affirm that an early unimodal OFS improves premature infants’ cardiorespiratory evolution and/or the acquisition of food autonomy, it does indicate an improved food efficiency during their first autonomous feedings.

## 1. Introduction

The transition from enteral to autonomous oral feeding is a fundamental concern for both parents and caregivers, since it is a determinant for the length of a preterm neonate’s hospital stay. Premature infants (<37 weeks GA) are at risk of feeding difficulties and have a delay in achieving oral food intake autonomy, especially in the context of prolonged artificial nutrition [1]. Oral feeding requires a coordination of sucking, swallowing, and breathing. This coordination is not mature until 32 weeks GA [2]. In a hospital setting, it is the coordination of sucking in conjunction with swallowing and respiratory control that will ultimately lead to the learning of oral feeding in a safe and effective manner [3].

A distinction is made between nutritive sucking (NS) when ingesting milk, and non-nutritive sucking (NNS) without the ingestion of liquid (nipple or dummy). NNS provides certain benefits that are not directly related to the acquisition of oral feeding [4,5]. These include promoting weight gain, producing an analgesic effect, reducing stress, stabilizing the child’s behavior and accelerating the progression of oral nutrition. Additionally, it is a good marker of the level of maturation of sucking itself. Factors that to help promote the maturation of NNS include oral stimulation through NNS or sensorimotor stimulation of the oral structures, performed during enteral feeding of clinically stable premature infants over 30 weeks GA [6].

Early interventions in NICU are essentially based on two main axes: traditional educational intervention strategies and neuroprotective strategies of various kinds. The NIDCAP program grew out of Brazelton’s work on newborn competence, which was continued by Ashbaugh [7]. It is an early developmental care program integrated into daily medical care and individualized through behavioral observations focused on the child and family. It aims to promote the harmonious development of the child in its various components: physiological, neurological, behavioral and relational and to improve the future quality of life. The QUALIN questionnaire was used in our study as a hetero-assessment of quality of life [8]. Research shows that prematurely born children’s development takes place more in a multi-sensory dynamic rather than single-sensory stimulation. Weekly parent-administered massage therapy, when combined with visual interactions between the parents and children, result in a more rapid acquisition of food autonomy. There is often a two-fold increase in breastfeeding rates at discharge, including reduced maternal stress, and a likely increase in breastfeeding efficiency [9,10,11], suggest that vagal stimulation resulting from combined tactile and kinesthetic stimulation allows an increase in gastric motility, which leads to an increase in weight gain. In addition, multisensory intervention (auditory, visual, tactile, and vestibular) stimulates brain development by modifying the cerebral structure of the child. This allows the maturation of food capacities by increasing the number of sucks and the average number of sucks per burst, after performing this intervention after 32 weeks of GA corrected age, in newborns between 29 and 34 weeks of age [12,13].

Finally, three studies compared multimodal and unimodal stimulation and suggested: (1) that a combined oral and tactile stimulation program would be more effective than a tactile stimulation program alone, for the duration of acquisition of food autonomy, weight gain, and length of hospital stay for premature infants [14]; (2) multimodal oral, tactile and kinesthetic stimulation would improve food efficiency compared to unimodal stimulation: oral or tactile/kinesthetic [15]; and (3) multimodal stimulation by NNS and auditory stimulation would shorten the transition period between autonomous enteral and oral feeding as compared to NNS alone [16]. This phenomenon could be explained by the synergistic effect of multimodal stimulation on food parameters.

In 2002, Fucile et al. reported that there was a more rapid acquisition of food autonomy after 10 days of unimodal oral stimulation, with no differences found on the duration of hospitalization [17]. Recently, a review of the Cochrane library [18] suggested a decrease in the transition time to autonomous oral feeding and a shorter hospital stay with NNS.

However, to our knowledge, there are no studies that have found any effect on the presence of cardio-respiratory events while improving food autonomy or weight gain. However, studies are methodologically heterogeneous, thus making it difficult to interpret the results. This calls into question the usefulness of these interventions in current practice.

Our study’s objectives were to evaluate if early NNS unimodal orofacial stimulation (OFS) reduces apnea–bradycardia and/or desaturations, and improves food autonomy in very premature infants.

## 2. Methods

### 2.1. Design

This study was monocentric, controlled, open, and randomized into two groups. Recruitment was prospective. The experimental group was represented by newborns receiving OFS associated with enteral nutrition, based on a program inspired by Fucile [6,16]. The control group corresponded to newborns having enteral feeding without receiving any OFS. Both groups received NNS from a pacifier.

### 2.2. Ethics Approval and Consent to Participate

The protocol was approved by the institutional ethics committee and by the French Protection Committee (N° IDRCB 2009-A01191-56, 1 January 2010), by the French National Institution of Pharmacovigilance (ANSM, Agence Nationale de Sécurité du Médicament et des produits de santé) (27 November 2009). Informed consent was obtained once the information leaflet was reviewed by the parents or legal representatives of the children. Participants were advised they could withdraw their consent at any time, and that the data would be kept confidential.

The ClinicalTrials.gov identifier is NCT01116765.

### 2.3. Sample

This centrally administered trial was conducted in a level 3 maternity unit in Marseille, France, and included neonates born between January 2013 and December 2014. In the study hospital, there are 2700 live births per year, 8% of which are preterm births (1.9% very preterm birth).

The inclusion criteria were:(1)Neonate born between 26 and 29 completed week GA;(2)Less or equal 33 week GA corrected age (CA);(3)Hospitalized in our neonatal unit;(4)Without neurological pathologies; confirmed by a normal transfontanellar ultrasound or grade 1 or 2 intraventricular hemorrhage [19], as well as a normal cerebral MRI (performed at corrected-term age) or type 1 to 6 brain abnormalities according to the modified Paneth classification [20] (in Paneth’s modified classification, six types of MRI are outlined: (1) normal MRI; (2) localized abnormality of white matter; (3) non-parenchymal hemorrhage: subependymal and intraventricular; (4) delay in myelination; (5) diffuse abnormality of white matter; and (6) other lesions: basal ganglia, cerebellum. MRI results were divided into two stages of severity: Group I with normal MRI or moderate abnormalities (Types 1, 2, 3, and 4) and Group II with severe abnormalities (Types 5 and 6));(5)Well-tolerated enteral feeding (without clinical symptoms suggestive of necrotizing enterocolitis defined by abdominal distension and/or increased gastric residuals (>20% of enteral feeding volume) and/or blood in stools (macro- or microscopic), greater than 100 mL/kg/day;(6)Without infectious pathologies making them clinically unstable (reactive C protein less than 7 mg/L);(7)Any CPAP (control positive airway pressure) for at least 48 h;(8)Had no congenital anomalies;(9)Both parents and their legal representatives agreed to participate in the study and signed an informed consent.

The exclusion criteria were: (1) children for whom one of the parents/legal guardians had withdrawn consent before the end of the study; (2) children with co-morbidity during the stimulation period requiring the sessions to be stopped (nosocomial infection and/or ulcerative-necrotizing enteritis and/or respiratory worsening requiring cessation of feeding and/or resumption of CPAP and/or an intensive care admission; (3) children transferred to another establishment before the end of the program; (4) children whose program sequence was not fully completed; and (5) deceased children.

The total duration of inclusion was set at 18 months.

In our NICU, the standard of care protocol is:-Initiation of enteral feeding: from the first few days of life, for all newborns;-Rate of increase in feeding;-Birth weight < 10th percentile, Weight < 1000 g, Preterm < 28 GA: start with digestive stimulation (same 20 mL/kg/d for 3–5 days); after digestive stimulation if well tolerated, increase enteral feeding by 20–25 mL/kg/d daily;-For eutrophic newborn and weight >1000 g: start at 20 mL/kg/d: if well tolerated, increase enteral feeding by 30–35 mL/kg/D daily.

The start of active feeding, for newborns born <33 weeks GA, is 33 weeks CA.

### 2.4. Intervention

The program included 10 consecutive days of OFS for the very preterm newborns, before 33 weeks GA corrected age. Twice a day, for 10 consecutive days, a 15 to 20 min sensorimotor tactile stimulation was performed 15 to 30 min before feeding by gastric tube. The caretaker performed careful handwashing and gloving prior to each session. The first 12 min of stimulation involved the cheeks, lips, gums and tongue, and the last 3 min consisted of suction stimulation (NNS).

The children were calm and awake in either their crib or incubator and placed in a semi-seated position. The pediatric nurse or parent supported the child’s head and assumed an enveloping and secure position. No stimulation was performed if the child was in deep sleep, or if they showed signs of discomfort or fatigue. This was evaluated by nurses who had been trained in NIDCAP development care and used the Newborn Individualized Developmental Care and Assessment Program (NIDCAP) grid [21,22].

### 2.5. Measurements

The primary endpoint was the assessment of apnea–bradycardia (with or without associated desaturations). Apnea–bradycardia with or without desaturation corresponded to a respiratory pause >10 s associated with bradycardia <80 beats per minute, associated, or not, with desaturation <80–85% in pulse oximetry. An isolated desaturation corresponded to an oxygen saturation <90%. The monitoring of the cardio-respiratory manifestations of the children of the two groups was carried out from their inclusion in the study to discharge from the neonatology department. The vital constants were assessed in real time or by retrospectively analyzing traces recorded by the scope. Cardio-respiratory events were recorded 30 min before, during, and three hours after each feeding.

The secondary objectives included outcomes variables (impact of early NNS unimodal OFS) were:-Evaluation of isolated desaturations during the learning period towards food autonomy with an analysis at four different times: (1) at the first autonomous feeding; (2) up to four independent feedings; (3) up to eight independent feedings on 2 consecutive days; and (4) and then all these autonomous feedings combined.-A validation of our OFS protocol by evaluating its tolerance and listing the cardio-respiratory manifestations (apneas, bradycardias, and desaturations at all feeds from day 1 to day 10 of the stimulation program, always 30 min before, during and in the hour after feeding).-The acquisition of food autonomy, which was defined when the newborn achieved eight independent feedings for two consecutive days.-The date of the first autonomous feeding.-The amount of milk ingested (mL/d).-Weight increase (g/kg/d).-The length of hospital stay.-Dietary monitoring at six months corrected age.-The quality of life of the child, assessed by the parents, using the QUALIN questionnaire [8], at 6 and 12 months of corrected age.

## 3. Statistical Analyses

Randomization was established using the OLAN procedure (SAS software), in order to divide the participants into two parallel and comparable groups. Data analyses were performed using SPSS version 20.0 software. The qualitative variables were presented in the form of percentages and the quantitative variables were in the form of mean ± standard deviation. The normality of the distributions of the quantitative data was systematically sought using the Shapiro–Wilks test. If several variables were not normally distributed, usual transformation techniques (logarithmic, etc.) or normalization algorithms (Blom or Tukey algorithms) were used to measure the interpretation of the results that resulted.

The statistical tests used were: The Student’s test or the non-parametric Mann–Whitney test for comparing means, the median test, Chi2 test or Fisher’s exact test for qualitative variables All tests were bilateral. The median and the mean were similar for all quantitative variables.

The number of subjects required was calculated for the main composite criterion: the number of apnea–bradycardias (with or without associated desaturations) and the number of desaturations. To highlight a decrease of at least two apnea–bradycardias or two desaturations during the four different times of autonomous feedings, it was evaluated with 21 patients per group with a risk of error of α = 0.05 and an expected power of 80%. The significance threshold was fixed at *p* = 0.05 in a bilateral situation.

The primary and secondary endpoint analyses were done on an intention-to-treat basis. The multivariate analyses consisted of modeling the primary and secondary endpoints, according to the two study groups, with parameters: gestational age, birth weight, and total ventilation time (invasive and non-invasive).

## 4. Results

### 4.1. Characteristic of Sample

There were 53 patients included and they were randomized. Of the 35 who were not excluded, 17 were in the experimental group with OFS and 18 were in the control group without OFS.

During the 12-month follow-up, eight patients were lost to follow-up: four from the experimental group and four from the control group (Figure 1).

There were 11 boys (64.7%) in the “with OFS” group and nine boys (50%) in the “without OFS” group. The mean term of birth was 28 weeks GA in the two groups. The average birth weight was 1101 g in the experimental group and 1047 g in the control group.

The baseline age at inclusion was 16.5 days (30 weeks GA and 2 days corrected age) in the experimental group, and 18 days, (30 weeks GA and 4 days corrected age) in the control group. There were no significant differences between the two groups, and both groups were comparable in terms of all their antenatal, neonatal, and postnatal characteristics (Table 1).

### 4.2. Primary Objective

The cardio-respiratory manifestations were significantly lower in the experimental group compared with the control group (respectively *n* = 102 vs. *n* = 61, *p* = 0.003) but the differences observed between the two groups were no longer significant (*p* = 0.39) after adjustment for gestational age, birth weight, and total duration of ventilation. There were no statistically significant differences between the two groups when detailing apnea–bradycardia with or without associated desaturations (Table 2).

### 4.3. Secondary Objectives

There were no statistically significant differences between the two groups concerning isolated desaturations during the four different times of the analysis (1, 4, and 8 autonomous feedings or all these feedings combined (Table 2)).

There appeared to be no intolerance to the OFS protocol. The weight gain in both groups appeared similar.

Although not significant (*p* = 0.70), the experimental group’s infants had a higher weight when discharged (2877 (±366) g) than the control group (2821 (±474) g). The total hospital stay length was shorter in the OFS group than the control group (respectively, 76.18 (±15.88) days vs. 78.72 (±17.67) days, *p* = 0.66) (Table 3).

The completion date for the eight autonomous feedings, continuing for 48 consecutive hours, which represented the date of acquisition of food autonomy, was earlier in the experimental group (corrected age of 36.41 (±1.06) weeks GA) than in the control group (36.78 (±1.44) weeks GA), but the difference was not significant (*p* = 0.40) (Figure 2).

Only the quantity of milk ingested during the first autonomous feeding was significantly higher in the experimental group than in the control group (respectively 34.11 mL (+/− 6.60) vs. 29.50 mL (+/− 5, 94), *p* = 0.04), in the univariate analysis. In the multivariate analysis with the adjustment for the parameters (gestational age, birth weight and total ventilation time), the observed difference between the two groups was no longer significant, however close to being significant (*p* = 0.07) (Appendix A).

A six-month nutritional follow-up concerning the during of breast feedings, achievement of diversification and the presence of a normal gag reflex, was comparable in the two groups. The infant’s quality of life at 6 and 12 months was evaluated using the QUALIN questionnaire for the parents and showed no significant difference between the two groups (Appendix A).

## 5. Discussion

Our study’s results suggest that early unimodal oral sensorimotor tactile stimulation does not significantly improve the cardio-respiratory stability or food autonomy of very premature babies.

No effect was found on the quantity of milk ingested (except at the first feeding), weight gain, or the length of hospital stay. The nutritional outcome at six months was unchanged, as was the quality of life of the child at 6 and 12 months, as assessed by the parents.

Similar to our study, a recent meta-analysis studied the effects of NNS in preterm infants under 37 weeks GA and found no significant effect on the decrease in cardio-respiratory events [23].

On the other hand, our results concerning food autonomy are not consistent with most of the literature. Numerous studies have suggested that the duration between the transition from enteral feeding to autonomous oral feeding was reduced through oral stimulation [24,25,26,27,28]. Our study showed a greater quantity of milk ingested during the first feeding which remained within the limits of significance. The average reduction time found in the different studies was 5 to 10 days. Rocha et al., in their randomized trial of 98 premature infants between 26 and 32 weeks GA found a more rapid 9-day acquisition of food autonomy [29]. Tian et al., in their meta-analysis of 855 newborns between 27 and 33 weeks of age, found a similar result, with a decrease of around four days [30]. Lessen et al. found that, for 20 premature newborns from 29 weeks GA, the decrease was five days [31], as in our study, the correlation was even more marked when the protocols used combined sensorimotor oral stimulation with NNS [32,33].

Finally, the results concerning the effect of oral stimulation on the length of hospital stays are heterogeneous [28,34]. For Rocha et al., as well as Tian et al., the significant reduction in the length of the hospital stay was observed to be between four to ten days. For Lessen et al., an effect of oral stimulation on the length of hospital stay was not found.

Other studies did not find any significant differences either on food autonomy or on the length of the hospital stay [34]. This is also the case in the study by Bache et al., which showed an increase in the rate of breastfeeding on discharge from the hospital, which was not the case in our study [35].

Our study’s strengths included its randomized nature, the presence of a comparable control group, the analysis carried out with an intention-to-treat basis, and the analysis of multiple parameters (secondary criteria). Strengths also included the frequently used OFS oral protocols, which resulted from Bruwier’s and Fucile’s work, and for which no signs of intolerance were noted. The study’s paramedical staff received OFS training by qualified professional staff and the inclusion of a neonatology speech therapist was an additional asset. We were able to avoid confounding factors that can modify orality by using multivariate analyses with adjustment for gestational age, birth weight, and total ventilation time. Finally, we were able to take into account the quality of life of children as assessed by the parents in the medium term.

Our study’s limitations were represented by the absence of a blind methodology as well as the unicentric nature of the trial. The large number of people lost to follow-up minimized the number of necessary subjects, and certain parameters at the limit of significance could have been improved. When the infant received oral stimulation by different personnel such as childcare workers or childcare assistants, the reproducibility of the sessions may not have been similar. For hygienic reasons, the personnel’s gloved finger used might have been a source of discomfort for the child. We were unable to measure the quantitative parameters of the milk suction rate transfer, or the quantity of milk ingested during the first five minutes, on the total milk consumption or even the nutritional efficiency. Studies having analyzed the sucking measurement parameters [36,37,38], note an improvement with oral stimulation, in particular the rate of milk transfer (quantity of milk ingested/duration of feeding in milliliters per minute), as well as the sucking frequency. This appears especially true since the quantity of milk ingested during first autonomous feed is at the limit of significance in multivariate analysis.

A major study weakness was the lack of details concerning OFS sessions which had no parent present. It can easily be assumed that when parents were present during the sessions, the newborn had conditions which enhanced their tolerance and well-being.

Finally, with improved developmental care for newborns, initiation to oral feeding should be done on the basis of personal observation for each newborn and not systematically according to their given gestational age. For individual adaptation, quantitative evaluation scores of nutritive suction make it possible to determine if the child is ready for this initiation to oral feeding [39].

## 6. Conclusions

Early non-nutritive OFS does not reduce cardio-respiratory events during the period of acquisition of food autonomy, nor does it reduce the duration of acquisition of food autonomy, but it may improve nutritional efficiency. Results on this subject remain controversial in the literature. Continued studies on the subject with a more robust methodology, and with a dynamic analysis of sucking measurements, multimodal stimulation seem appropriate.

## Figures and Tables

**Figure 1 children-08-01188-f001:**
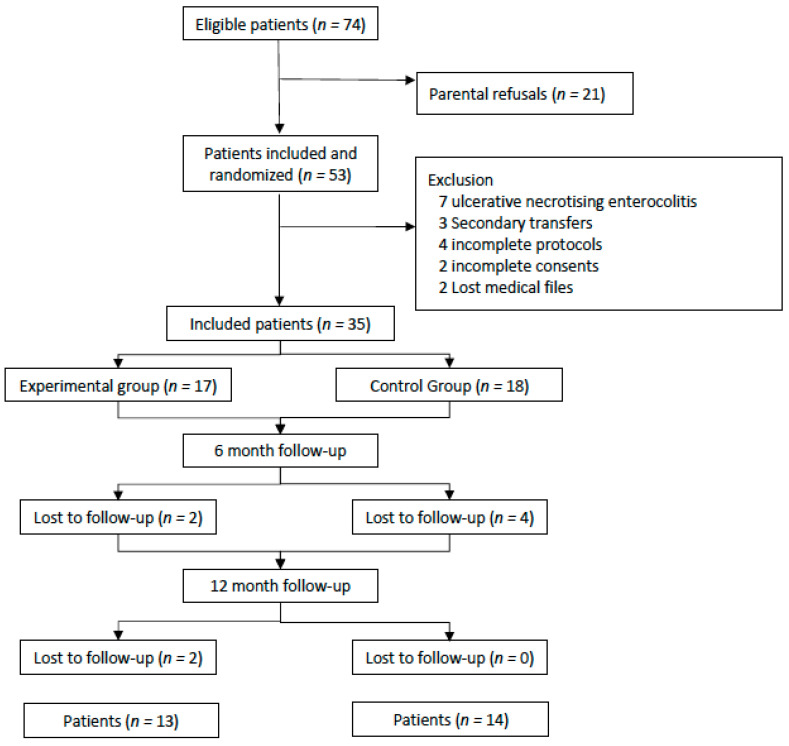
Flow chart.

**Figure 2 children-08-01188-f002:**
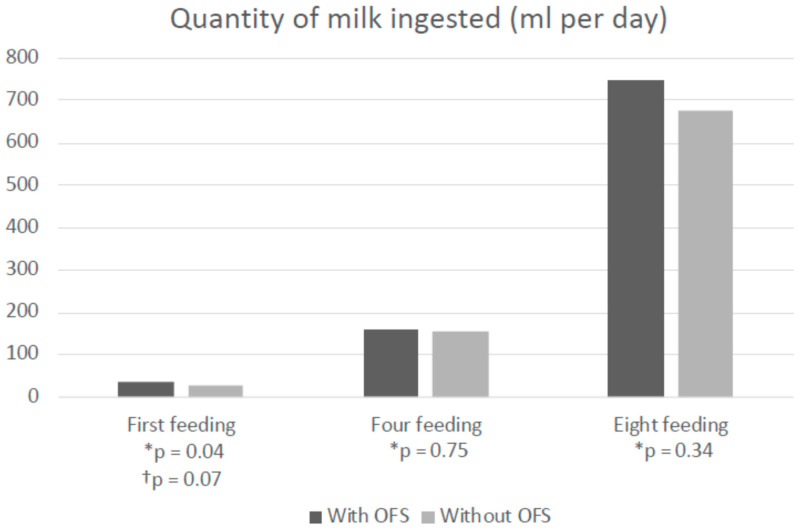
Amount of milk ingested during the 1st, 4th and 8th independent feeding (ml), in the experimental (OFS *n* = 17) vs. control group (without OFS, *n* = 18). *p*: value of the difference between groups with available data; * Value of *p* in univariate analysis. † Value of *p* in multivariate analysis including Gestational Age, Birth Weight and total duration of ventilation. *p* < 0.05: statistically significant difference.

**Table 1 children-08-01188-t001:** Characteristics of study population.

Characteristics	With OFS (*n* = 17)	Without OFS (*n* = 18)	*p*
**ANTENATAL**			
Maternal age (years), mean (±SD)	31.94 (±5.2)	28.89 (±7.1)	0.15
Twin pregnancies, *n* (%)	4 (23.5)	4 (22.2)	1.00
Pregnancies with pathology, *n* (%)	6 (35.3)	4 (22.2)	1.00
Intrauterine growth retardation, *n* (%)	0 (0)	2 (11.1)	0.48
Cesarean births, *n* (%)	12 (70.6)	14 (77.8)	0.47
NEONATAL			
Weeks GA (GA), average (±SD)	27.98 (±1.3)	28.07 (±1.3)	0.55
Males, *n* (%)	11 (64.7)	9 (50)	0.50
Average birth weight (g), mean (±SD)	1001.76 (±224.7)	1046.94 (±181.8)	0.56
SGA, *n* (%)	1 (5.9)	3 (16.7)	0.34
**POSTNATAL**			
Week corrected gestational age at inclusion (GA), mean (±SD)	30.37 (±1.1)	30.60 (±1.0)	0.41
Age at inclusion (days), mean (±SD)	16.6 (±9.8)	17.8 (±11.5)	0.92
Parenteral feeding time (days), mean (±SD)	21.1 (±9.6)	23.7 (±9.6)	0.42
Total ventilation time (days), mean (±SD)	22.9 (±13,4)	21 (±12)	0.73
Total invasive ventilation time (days mean (±SD)	5.7 (±6.9)	4 (±3.1)	0.81
Non-invasive ventilation time (days), mean (±SD)	18.2 (±8.4)	17.8 (±10.6)	0.74
Weight at inclusions (grams), mean (±SD)	1471 (±196)	1449 (±274)	0.79

OFS: orofacial stimulation; GA: gestational age; SGA: small for gestational age. Data are presented as *n* (%) unless stated differently. *p* value of the difference between groups with available data, *p* < 0.05: statistically significant difference.

**Table 2 children-08-01188-t002:** Number of apnea–bradycardia (with or without desaturations), isolated desaturations and all cardiorespiratory, events combined, in experimental versus control group, during transition to food autonomy.

With OFS (*n* = 17)	Without OFS (*n* = 18)	*p* *	ß	*p* ^†^
Number of apnea–bradycardia (±desaturations) per patient, mean (±SD)				
0.59 (±0.80)	0.33 (±0.77)	0.34		
0.82 (±1.29)	1.18 (±1.55)	0.48		
1.94 (±2.77)	3.89 (±4.97)	0.16		
3.24 (±3.55)	5.33 (±6.04)	0.22	1.21	0.50
Number of isolated desaturations per patient, mean (±SD)				
0 (±0)	0.17 (±0.38)	0.08		
0.59 (±1.37)	0.41 (±1.70)	0.74		
0.81 (±1.76)	1.17 (±2.20)	0.61		
1.35 (±2.71)	1.72 (±2.54)	0.68	0.55	0.56
Number of feeds with cardiorespiratory events (all combined)/number of feeds without cardiorespiratory events				
61/271	102/265	0.003 ^‡^	1.53	0.39

OFS: orofacial stimulation; SD: standard deviation. *p* value of the difference between the groups with available data; * value of *p* in univariate analysis; ^†^ value of *p* in multivariate analysis including the parameters: gestational age, birth weight and total duration of ventilation; ß: correlation coefficient in multivariate analysis; ^‡^  *p* < 0.05: statistically significant difference.

**Table 3 children-08-01188-t003:** Evaluation: tolerance of OFS protocol (tolerance day 1 to day 10).

	With OFS (*n* = 17)	Without OFS (*n* = 18)	*p* *	ß	*p* ^†^
Number of apneas	3.13 (±6.15)	3.17 (±5.85)	0.99	−0.69	0.65
Number of bradycardias	16.41 (±15.58)	21.28 (±17.37)	0.39	8.26	0.15
Number of saturations	8.24 (±12.16)	9.67 (±10.02)	0.71	2.31	0.53
Weight difference between day 1 and day 10 (g)	240 (±94)	237 (±87)	0.92	−17.1	0.57
Discharge weight (g)	2877 (±366)	2821 (±474)	0.70		
Total hospital stay (days)	76.18 (±15.88)	78.72 (±17.67)	0.66		

Data are presented as mean (±SD). Mean number of apneas, bradycardias and desaturations, per patient, during the 10 days of OFS: difference in weight between the first and tenth day of the OFS protocol; weight at discharge; total length of hospital stay in the experimental vs. the control group. OFS: orofacial stimulation; SD: standard deviation. *p* value of the difference between the groups with available data, *p* < 0.05: statistically significant difference. * Value of *p* in univariate analysis. ^†^ Value of *p* in multivariate analysis including the parameters: gestational age, birth weight and total duration of ventilation. ß: correlation coefficient of the multivariate analyses.

## Data Availability

The datasets that generated and/or analyzed during the current study are not publicly available due to the data belongs to the Assistance Publique Hopitaux de Marseille. However, datasets are available from the sponsor (promotion.interne@ap-hm.fr) on reasonable request and after sign a contract pertaining to the provision of data and/or results.

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
