# Peer review of "Cardio-Respiratory Events and Food Autonomy Responses to Early Uni-Modal Orofacial Stimulation in Very Premature Babies: A Randomized, Controlled Study"

_children, 2021, doi:10.3390/children8121188_

Round 1

Reviewer 1 Report

Many thanks for this paper, for which I have a few commenst, detailed below

Review:

This is a report of a randomized controlled study of an intervention (oro-facial stimulation, OFS) aimed at

  • reducing cardiorespiratory events during feeding in very preterm neonates
  • improving feeding competence in preterm patients, as measured with ingested volume during first feed, time to first autonomous feed, and time to full oral feeding

Importance of the subject: The feeding of very preterm infants remains a difficult question, as feeding difficulties are responsible for longer hospitalisation and therefore higher costs, and may put a burden on parent-child relationships. Research has not yet identified evidence-based interventions effective to shorten time to full oral feeding.

General appreciation: This is a well constructed paper reporting on an open, not blinded randomized control trial, with several limits which are given in the discussion. The article is easy to follow, but the quality of the writing and of the English varies, the sections seem to have been written by different co-authors.

Introduction: Short literature review on the subject. This section needs editing for the English and for the content. For example:

Line 36: premature < 37 weeks is not the subject of this reserach

Line 37-80: early oral disorder…: not clear what is meant

Line 41: “preparation…”: not clear what is meant

Line 45: NNS offers: rephrase, not clear

Line 53: no difference on/for the duration, rather than over

Line 56: no study found--- rephrase

Methods: generally well written, section clear

Design: line 66: rephrase, give the intervention (OFS) and control. Sham procedure?

Sample:

At some point, describe the standard of care protocol in your institution, when do you start enteral feeding, rate of increase, start of oral feeding etc, use of milk or sugar for pain relief.

Line 82: transfontanellar ultrasound

Line 83: MRI done when? what lesions, reference is not correct,  describe

Line 84: how do you define well tolerated?

Intervention:

Line 98: starting when, based on which cues? Are these 30 weekers who are awake 30 minutes before a meal? Did you give any milk in the mouth?

Setting: line 114: 2700 births, 8% very preterm (<32 weeks), is this right? This would make an n of 432 during the study period

Measurement:

First paragraph should be rephrased, these timepoints are difficult to understand. Each one is included in the next if I understood well?

Secondary outcomes:

Line 131-133: so cardiopulmonary tolerance of OFS before the start of any oral feeding?

Line 138: amount by feed, day? Percentage of feed prescribed?

Line 139: weight increase by which units?

Statistical analysis: very clear

Results: well presented, English needs a bit of proof-reading  (example: line 193: we noted… better to say: there was no statistical…). Time points should be presented differently, see section “measurements”.

Discussion: well written, easy to follow, thank you.

Author Response

  • Reviewer 1:

Dear Doctor, Dear Reviewer,

We thank you so much for your valuable and appreciated inputs in our article. The interest you showed in this regard encourages us to continue our research.

Respectfully

  1. Introduction: Short literature review on the subject. This section needs editing for the English and for the content. For example:

Line 36: premature < 37 weeks is not the subject of this research

The inclusion criteria were : 1) neonate born between 26 and 29 completed Week’s GA, 2) less or equal 33 Week’s GA corrected age (CA), 3) hospitalized in our neonatal unit; so preterm infant was the subject of this research. Thank you.

Line 37-80: early oral disorder…: not clear what is meant

Thank you, we have modified the sentence: Premature infants (<37 weeks GA) are at risk for feeding difficulties and have a delay in achieving oral food intake autonomy, especially in the context of prolonged artificial nutrition [1].

Line 41: “preparation…”: not clear what is meant

Thank you for your comment: “that will ultimately lead to learning of oral feeding”

Line 45: NNS offers: rephrase, not clear

Thank you for your comment: “NNS provides certain benefits that are not directly related to the acquisition of oral feeding”

Line 53: no difference on/for the duration, rather than over; Line 56: no study found--- rephrase

Thank you for your comment. The text and this paragraph have been rewritten.

  1. Methods: generally well written, section clear

Design: line 66: rephrase, give the intervention (OFS) and control. Sham procedure?

  1. Sample: At some point, describe the standard of care protocol in your institution, when do you start enteral feeding, rate of increase, start of oral feeding etc, use of milk or sugar for pain relief.

Thank you for your comment

- Initiation of enteral feeding: From the first few days of life, for all newborns.

- Rate of increase in feeding:

- Birth weight < 10th percentile, Weight < 1000g, Preterm < 28GA: start with digestive stimulation (same 20 ml/kg/d for 3-5 days); after digestive stimulation if well tolerated, increase enteral feeding by 20-25ml/kg/d daily.

- For eutrophic newborn and weight >1000g: start at 20ml/kg/d: if well tolerated, increase enteral feeding by 30-35mL/kg/D daily.

The start of active feeding, for newborns born <33 weeks GA, is 33 weeks CA.

Line 82: transfontanellar ultrasound

Thank you, correction done.

Line 83: MRI done when? what lesions, reference is not correct, describe

Thank you for your comment.

Brain MRI performed at corrected-term age.

The preterm infants’ lesions were classified following Paneth’s model according to their anatomical distribution. In Paneth’s modified classification, six types of MRI are outlined: 1) normal MRI; 2) localized abnormality of white matter; 3) non-parenchymal hemorrhage: subependymal and intraventricular; 4) delay in myelination; 5) diffuse abnormality of white matter; and 6) other lesions: basal ganglia, cerebellum. MRI results were divided into two stages of severity: Group I with normal MRI or moderate abnormalities (Types 1, 2, 3, and 4); and Group II with severe abnormalities (Types 5 and 6).

Line 84: how do you define well tolerated?

Thank you for your pertinent comment, we have been modified the sentence: enteral feeding is considered to be well tolerated in the absence of clinical symptoms suggestive of necrotizing enterocolitis defined by: abdominal distension and/or increased gastric residuals (> 20% of enteral feeding volume) and/or blood in stools (macro- or microscopic)

  1. Intervention: Line 98: starting when, based on which cues? Are these 30 weekers who are awake 30 minutes before a meal? Did you give any milk in the mouth?

Included are neonate born between 26 and 29 completed Week’s GA and the intervention is done on average at 30 weeks CA (16-18 days post birth) because enteral feeding protocol allows to feed at 100 ml/kg/d at the second week of life (see above in correction).

  1. Setting: line 114: 2700 births, 8% very preterm (<32 weeks), is this right? This would make an n of 432 during the study period

We are sorry, it was a mistake. 8% of children are born preterm, of which 15% are very preterm birth (1.9% of births)

  1. Measurement: First paragraph should be rephrased, these timepoints are difficult to understand. Each one is included in the next if I understood well?

Thank you for your comment, the text and this paragraph have been rewritten.

Measurements

The primary endpoint was composite and included the assessment of apnea-bradycardia (with or without associated desaturations). Apnea-bradycardia with or without desaturation corresponded to a respiratory pause >10 seconds associated with bradycardia <80 beats per minute, associated, or not, with desaturation <80-85% on the pulse oximetry. An isolated desaturation corresponded to an oxygen saturation <90%. The monitoring of the cardio-respiratory manifestations of the children of the two groups was carried out from their inclusion in the study to their discharge from the neonatology department. The vital constants were assessed in real time or by retrospectively analyzing traces recorded by the scope. Cardio-respiratory manifestations were recorded 30 minutes before, during, and three hours after each feeding. It additionally included the evaluation of isolated desaturations during the learning period towards food autonomy with an analysis at four different times: 1) at the first autonomous feeding, 2) up to four independent feedings, 3) up to eight independent feedings on 2 consecutive days, and 4) and then all these autonomous feedings combined.

Apnea-bradycardia with or without desaturation corresponded to a respiratory pause >10 seconds associated with bradycardia <80 beats per minute, associated, or not, with desaturation <80-85% on the pulse oximetry. An isolated desaturation corresponded to an oxygen saturation <90%. The monitoring of the cardio-respiratory manifestations of the children of the two groups was carried out from their inclusion in the study to their discharge from the neonatology department. The vital constants were assessed in real time or by retrospectively analyzing traces recorded by the scope. Cardio-respiratory events were recorded 30 minutes before, during, and three hours after each feeding.

The secondary outcomes included

- It additionally included the evaluation of isolated desaturations during the learning period towards food autonomy with an analysis at four different times: 1) at the first autonomous feeding, 2) up to four independent feedings, 3) up to eight independent feedings on 2 consecutive days, and 4) and then all these autonomous feedings combined.

- A validation of our OFS protocol by evaluating its tolerance and listing the cardio-respiratory manifestations (apneas, bradycardias and desaturations at all feeds from D1 to D10 of the stimulation program, always 30 minutes before, during and in the hour after feeding).

  1. Secondary outcomes:Line 131-133: so cardiopulmonary tolerance of OFS before the start of any oral feeding?

YES

Line 138: amount by feed, day? Percentage of feed prescribed?;

Thank you, we have specified it.

Line 139: weight increase by which units?

We have modified according to your comments.

  1. Results: well presented, English needs a bit of proof-reading  (example: line 193: we noted… better to say: there was no statistical…). Time points should be presented differently, see section “measurements”.

Thank you. Correction done.

Reviewer 2 Report

The manuscript Chidren 1456643 entitled "Cardio-respiratory events and food autonomy responses to early uni-modal orofacial stimulation in very premature infants: a randomized, controlled study" is a case-control design study that aims to establish the association between unimodal orofacial stimulation intervention and decreased cardiorespiratory events and improved food autonomy in preterm infants. 

The manuscript presented is pertinent, but requires some improvements to improve its quality, to establish a hypothesis of association, in accordance with the design of the study as a case-control, presenting the selection and description of the cases and controls in a more precise and orderly manner and elaborating the conclusions obtained.  

For these reasons, a major revision of the present version is recommended. The following specific comments are provided:

  • ABSTRACT: modify according to the recommendations offered in each of the sections.

1- INTRODUCTION:

The introduction gives a brief description of the object of study.

Due to the existence of different interventions or models of oral stimulation, although it includes a brief explanation of Fucile's model as a theoretical model, it is considered necessary to include/provide a definition of unimodal oral (sensorimotor) stimulation. It is also recommended to include a brief description of the Developmental Care and Assessment Program [NIDCAP] mentioned in the next section and the QUALIN questionnaire.

It is suggested to include in the introduction section previous studies that allow the limitation/determination of the state of the question and the justification of the study.

In relation to the objective, we recommend rewriting the objective with measurable verbs; and transforming the sentence: "Our study's objectives were to demonstrate that early NNS unimodal OroFacial Stim-60 ulation (OFS) reduces apnea-bradycardia and/or desaturations, and improves food autonomy in very premature infants born between 26 and 29 Weeks GA." formulated as the hypothesis of association.

  1. METHODS:

In general terms, the definition of cases and controls needs to be formulated more specifically and precisely, as well as the case-control selection procedure.

- Sample:

- Line 87: replace number 8 in the sequence by number 9, (number 8 is repeated).

- It is recommended to include the content of the Setting section (L:111-114) at the beginning of the participants section, prior to the description of the inclusion and exclusion criteria.

- Setting: it is suggested to include this section in the Sample section.

- Measurements: Variables are presented as primary endpoint (L116) and secondary outcomes included (L130).

It is recommended that they be presented by unifying the terms, primary and secondary variables, adding that the secondary variables are outcome variables. The use of terms as they are currently used may lead to confusion.

- Statistical analyses:

- Add the version of SPSS used.

- Indicate which variables have a normal distribution and which do not (and the p-value).

Although all the requirements attributable to this section are made explicit throughout this Methods section, its presentation requires a more detailed and orderly description of the variables and their analysis.

  1. RESULTS

These are presented with reference to "primary objective" and "secondary objective" but these terms have not been made explicit previously in the manuscript, including secondary objectives results related to secondary variables.

It is advisable to indicate the statistic or test used to establish significant differences between groups accompanying the results tables.  For example

- In L: 219-222, "The completion date for the eight autonomous feedings, continuing for 48 consecutive-219 tive hours, which represented the date of acquisition of food autonomy, was earlier in the 220 experimental group (corrected age of 36.41 (± 1.06) Weeks GA) than in the control group 221 (36.78 (± 1.44) Weeks GA), but the difference was not significant (p=0.40) What is the test or statistic used in this analysis?

- In L; 232-234, what is the test performed to establish that the differences are not significant between the two groups?

In addition to the results regarding the differences between groups and correlations presented, the determination of the Odds Ratio to measure the frequency of exposure would substantially improve the manuscript, complete the results, and allow the hypothesis of association to be accepted or rejected. The Odds Ratio is considered a key analytical parameter for case-control studies. 

  1. DISCUSSION:

Part of the discussion (298-318) can be considered state of the art, so its inclusion in the introduction is recommended, reserving the results of previous studies as part of the discussion with the results obtained in the study.

  1. CONCLUSION: a section on conclusions has not been prepared. It is recommended.

Author Response

  • Reviewer 2:

Dear Doctor, Dear Reviewer,

We thank you so much for your efficient and pertinent comments on our article: “Cardio-respiratory events and food autonomy responses to early uni-modal orofacial stimulation in very premature babies : a randomized, controlled study”.

There were very efficient and pertinent. We took them into consideration and have made changes accordingly.

We hope the changes we made will satisfy you.

Responses

1- INTRODUCTION:

The introduction gives a brief description of the object of study.

Due to the existence of different interventions or models of oral stimulation, although it includes a brief explanation of Fucile's model as a theoretical model, it is considered necessary to include/provide a definition of unimodal oral (sensorimotor) stimulation. It is also recommended to include a brief description of the Developmental Care and Assessment Program [NIDCAP] mentioned in the next section and the QUALIN questionnaire.

It is suggested to include in the introduction section previous studies that allow the limitation/determination of the state of the question and the justification of the study.

In relation to the objective, we recommend rewriting the objective with measurable verbs; and transforming the sentence: "Our study's objectives were to demonstrate that early NNS unimodal OroFacial Stim-60 ulation (OFS) reduces apnea-bradycardia and/or desaturations, and improves food autonomy in very premature infants born between 26 and 29 Weeks GA." formulated as the hypothesis of association.

Thanks, we have rewritten the introduction according to the comments

  1. METHODS:

In general terms, the definition of cases and controls needs to be formulated more specifically and precisely, as well as the case-control selection procedure.

- Sample:

- Line 87: replace number 8 in the sequence by number 9, (number 8 is repeated).

Thank you, correction done.

- It is recommended to include the content of the Setting section (L:111-114) at the beginning of the participants section, prior to the description of the inclusion and exclusion criteria. Setting: it is suggested to include this section in the Sample section.

Thank you, done.

- Measurements: Variables are presented as primary endpoint (L116) and secondary outcomes included (L130); It is recommended that they be presented by unifying the terms, primary and secondary variables, adding that the secondary variables are outcome variables. The use of terms as they are currently used may lead to confusion.

Thank you, we have clarified.

- Statistical analyses:

- Add the version of SPSS used.

- Indicate which variables have a normal distribution and which do not (and the p-value).

Although all the requirements attributable to this section are made explicit throughout this Methods section, its presentation requires a more detailed and orderly description of the variables and their analysis.

Thank you for your comment

  1. RESULTS

These are presented with reference to "primary objective" and "secondary objective" but these terms have not been made explicit previously in the manuscript, including secondary objectives results related to secondary variables.

Thank you, the results section has been modified, concerning primary and secondary objective.

It is advisable to indicate the statistic or test used to establish significant differences between groups accompanying the results tables.  For example

- In L: 219-222, "The completion date for the eight autonomous feedings, continuing for 48 consecutive-219 tive hours, which represented the date of acquisition of food autonomy, was earlier in the 220 experimental group (corrected age of 36.41 (± 1.06) Weeks GA) than in the control group 221 (36.78 (± 1.44) Weeks GA), but the difference was not significant (p=0.40) What is the test or statistic used in this analysis?

Thank you for your comment; The Student test or the non-parametric Mann-Whitney test for the median test (in L. 153-154)

- In L; 232-234, what is the test performed to establish that the differences are not significant between the two groups?

We had already specified in the legend of the supplemental table 1: p: value of the difference between the groups with available data, obtained by χ2 test. Thank you.

- In addition to the results regarding the differences between groups and correlations presented, the determination of the Odds Ratio to measure the frequency of exposure would substantially improve the manuscript, complete the results, and allow the hypothesis of association to be accepted or rejected. The Odds Ratio is considered a key analytical parameter for case-control studies. 

Correlation works with continuous data/measurement e.g weight, height, exact age etc. as mentioned by Michael, correlation looks at how much one of the variables is explained by the other variable. Like: How much is weight explained by someones height? It doesn't work well with categorical data. Thank you for your comment.

  1. DISCUSSION:

Part of the discussion (298-318) can be considered state of the art, so its inclusion in the introduction is recommended, reserving the results of previous studies as part of the discussion with the results obtained in the study.

We have taken your comment into account. Thank you.

  1. CONCLUSION: a section on conclusions has not been prepared. It is recommended.

We agree with you; a conclusion has been prepared.

Round 2

Reviewer 2 Report

Although most of the aspects mentioned in the introduction have been modified, a brief description of the Developmental Assessment and Care Program [NIDCAP] mentioned in the next section and the QUALIN questionnaire are still not reflected in the introduction.

L 84: It is recommended to state the objective, in measurable verbs; substituting the verb "to demonstrate" .

Author Response

Authors’ Response to Editor - Reviewers:

Thank you for your letter and for the reviewers’ comments on our manuscript entitled “Cardio-respiratory events and food autonomy responses to early uni-modal orofacial stimulation in very premature babies: a randomized, controlled study”. All of these comments were very helpful for revising and improving our paper. We have studied these comments carefully and have made corresponding corrections that we hope will meet with your approval. The changes in the revised manuscript are marked in red. The responses to the reviewers’ comments are provided below.

We would like to express our great appreciation to you and the reviewers for the comments on our paper.

Kind regards,

  • Reviewer 1:

Dear Doctor, Dear Reviewer,

We thank you so much for your valuable and appreciated inputs in our article. The interest you showed in this regard encourages us to continue our research.

Respectfully

Responses:

Although most of the aspects mentioned in the introduction have been modified, a brief description of the Developmental Assessment and Care Program [NIDCAP] mentioned in the next section and the QUALIN questionnaire are still not reflected in the introduction.

Thank you for your pertinent comment; we have been modified this section:

Early interventions in NICU are essentially based on two main axes: traditional edu-cational intervention strategies and neuroprotective strategies of various kinds. The NIDCAP program grew out of Brazelton's work on newborn competence, which was con-tinued by H. Ashbaugh [7]. It is an early developmental care program integrated into daily medical care and individualized through behavioral observations focused on the child and family. It aims to "promote the harmonious development of the child in its various components: physiological, neurological, behavioral and relational" and to improve the future quality of life. The QUALIN questionnaire was used in our study as a hetero-assessment of quality of life [8]. 

L 84: It is recommended to state the objective, in measurable verbs; substituting the verb "to demonstrate".

Thank you, we changed the word.